# Can Additional Patient Information Improve the Diagnostic Performance of Deep Learning for the Interpretation of Knee Osteoarthritis Severity

**DOI:** 10.3390/jcm9103341

**Published:** 2020-10-18

**Authors:** Dong Hyun Kim, Kyong Joon Lee, Dongjun Choi, Jae Ik Lee, Han Gyeol Choi, Yong Seuk Lee

**Affiliations:** 1Department of Orthopedic Surgery, Seoul National University College of Medicine, Seoul National University Bundang Hospital, Seoul 03080, Korea; osdrkdh@gmail.com (D.H.K.); jaeik15@gmail.com (J.I.L.); meinmed87@naver.com (H.G.C.); 2Department of Orthopaedic Surgery, Gwangmyeong 21st Century Hospital, Gyeonggi-do 14100, Korea; 3Department of Radiology, Seoul National University College of Medicine, Seoul National University Bundang Hospital, Seoul 03080, Korea; kjoon31@gmail.com (K.J.L.); chzze4582@gmail.com (D.C.)

**Keywords:** knee, osteoarthritis, diagnosing, deep learning, performance

## Abstract

The study compares the diagnostic performance of deep learning (DL) with that of the former radiologist reading of the Kellgren–Lawrence (KL) grade and evaluates whether additional patient data can improve the diagnostic performance of DL. From March 2003 to February 2017, 3000 patients with 4366 knee AP radiographs were randomly selected. DL was trained using knee images and clinical information in two stages. In the first stage, DL was trained only with images and then in the second stage, it was trained with image data and clinical information. In the test set of image data, the areas under the receiver operating characteristic curve (AUC)s of the DL algorithm in diagnosing KL 0 to KL 4 were 0.91 (95% confidence interval (CI), 0.88–0.95), 0.80 (95% CI, 0.76–0.84), 0.69 (95% CI, 0.64–0.73), 0.86 (95% CI, 0.83–0.89), and 0.96 (95% CI, 0.94–0.98), respectively. In the test set with image data and additional patient information, the AUCs of the DL algorithm in diagnosing KL 0 to KL 4 were 0.97 (95% confidence interval (CI), 0.71–0.74), 0.85 (95% CI, 0.80–0.86), 0.75 (95% CI, 0.66–0.73), 0.86 (95% CI, 0.79–0.85), and 0.95 (95% CI, 0.91–0.97), respectively. The diagnostic performance of image data with additional patient information showed a statistically significantly higher AUC than image data alone in diagnosing KL 0, 1, and 2 (*p*-values were 0.008, 0.020, and 0.027, respectively).The diagnostic performance of DL was comparable to that of the former radiologist reading of the knee osteoarthritis KL grade. Additional patient information improved DL diagnosis in interpreting early knee osteoarthritis.

## 1. Introduction

Osteoarthritis (OA) is the most common musculoskeletal disorder that involves inflammation and major structural changes of the joint [1]. This results in irreversible damage to the joint cartilage and bony structures [2]. The prevalence and economic burden of OA is very high; specifically, lower extremity OA is the eleventh highest global disability, with prevalence rising with age [1]. Moreover, it has been shown that the prevalence of symptomatic knee OA is especially high in Asians, with an estimated prevalence of 38% among those older than 65 years [3]. Possible risk factors of knee OA include age, obesity, gender (i.e., female), repetitive knee trauma, and life style, i.e., frequent kneeling [4]. The current gold standard for screening of OA is plain radiographic evaluation due to its relative safety, availability, and cost-efficiency [5,6]. The Kellgren–Lawrence (KL) grading system is the most commonly used knee OA severity grading scale. It divides knee OA into five severity grades, from 0 to 4 [7]. However, some previous studies have asserted that the KL grading system may be too ambiguous, due to disagreements between intra- and inter-rater reliability of quadratic Kappa, ranging from 0.56 to 0.67 [8,9,10]. These kinds of disagreements make the diagnosis of knee OA challenging and unreliable.

Deep learning (DL) using convolutional neural network (CNN) is an emerging technology. It has been recognized for its strengths in image classification, and as such, implementation of DL in diagnostic medicine has been heavily investigated, including the diagnosis of maxillary sinusitis with conventional radiography, detection of osteonecrosis of the femoral head with digital radiography, detection of moyamoya disease in plain skull radiography, and diagnosis of the severity of knee OA from plain radiographs [6,11,12,13]. To date, many studies have strived to improve the diagnostic performance, but to the best of our knowledge, they have mostly focused on using only radiologic data [14].

However, clinicians usually use various types of patient information to diagnose OA, determine the treatment modality, and predict the prognosis. Patient demographic data, additional radiologic data, such as alignment, and metabolic data, such as combined morbidity, can all be helpful in making the diagnosis. It was assumed that a combination of patient information and radiologic data in DL would increase the accuracy of diagnosing OA. However, to date, whether this was possible remained uncertain.

Therefore, this study intended to increase the performance of DL in diagnosing OA by combining additional patient information and radiologic data compared with radiologic data alone in DL. Hence, the purpose of this study was twofold: (1) to compare the diagnostic performance of DL with radiologic data alone compared with that of the former radiologist reading of the knee OA KL grade; and (2) to evaluate whether combining additional patient information (demographic, alignment, and metabolic data) with radiologic data in DL can improve diagnostic performance. Hypotheses of this study were as follows: (1) the diagnostic performance of DL would be unsatisfactory when compared with that of the former radiologist reading of knee OA; and (2) additional patient information would improve diagnostic performance of DL.

## 2. Materials and Methods

### 2.1. Dataset

From March 2003 to February 2017, a total of 72,258 patients suffering from knee pain and who subsequently underwent standing knee anteroposterior (AP) radiograph with formal reading of KL grade system were enrolled in this retrospective cohort study. Among them, 3000 patients with 4366 knee AP radiographs were randomly selected using stratified random sampling [15]. To avoid cluster effect between multiple radiographs in a single patient, only the initial knee AP radiograph was used.

The additional patient information, which might affect OA progression, was gathered from the clinical data warehouse (CDW). This information was categorized as demographic, alignment, and metabolic data. Age, gender, and body mass index (BMI) were included in the demographic data. The weight-bearing line (WBL) ratio was evaluated as a radiologic data. The WBL ratio was calculated as the percentage of the crossing point of the mechanical axis, from the medial edge of the tibial plateau to the entire width of the tibial plateau. The metabolic data were the history of diabetic mellitus (DM) and hypertension (HTN).

The image data and data regarding factors affecting OA were divided into training, validation, and test sets based on the ratio of KL grade. The training and validation set was divided by 9:1 ratio in proportion to KL grade (Table 1) [11,16].

All test set images were after July 2016.

The institutional review boards approved this study. The requirement for informed consent was waived due to the retrospective nature of this study and the use of anonymized patient data.

### 2.2. Labeling

All anonymized Digital Imaging and Communications in Medicine (DICOM) files of the enrolled patients were downloaded from the picture archiving and communication system (PACS) and used. All DICOM files were matched with radiologists’ former reading of KL grade.

All radiographs were labeled according to the semi-quantitative KL grade, which was divided into five categories; KL-0 (no presence of OA changes), KL-1 (doubtful narrowing of joint space with possible osteophyte formation), KL-2 (possible narrowing of the joint space with definite osteophyte formation), KL-3 (definite narrowing of joint space, moderate osteophyte formation, some sclerosis, and possible deformity of bony ends), and KL-4 (large osteophyte formation, severe narrowing of the joint space with marked sclerosis, and definite deformity of bone ends) [7,17].

### 2.3. DL Algorithm

In our study, DL algorithm was applied to a knee x-ray without cropping the joint area [14]. In the image of both knees, the right and left knees were cropped respectively. The right knee was cropped to 280 × 224 mm with a height of 0.5 and width of 0.25 in the knee image. The left knee was cropped to the same size as the right knee, with a height of 0.5 and a width of 0.75. The cropped knee was resized to 320 × 256 pixels. For data augmentation, horizontal or vertical shift and horizontal flip were applied to the training set. Pydicom library (version 1.2.0, Python software Foundation, Wilmington, DE, USA) was used for image processing in DICOM format.

We performed the training on CUDA/cuDNN (versions 9.0 and 7.4, respectively) and TensorFlow library (version 1.12, Google Brain Team, Mountain View, CA, USA) for graphic processing unit acceleration on a Linux operating system (Ubuntu 16.04, Canonical Ltd., London, UK).

Knee images and clinical information were trained by DL algorithms in two stages. In the first stage, a convolutional neural network was constructed by stacking six squeeze-and-excitation ResNet (SE-ResNet) modules to train only images [18]. The number of features in the last SE-ResNet module was 5, and after this block, Log-Sum-Exp pooling and Softmax activation function were sequentially applied to predict the KL grade, which ranged from 0 to 4. Xavier initialization was used as the initial value of parameters. In training the network, the learning rate, decay rate, and decay step were set to 0.01, 0.94, and 5000, respectively. The mini-batch size was set to six. Cross-entropy loss was minimized by applying the RMS Prop optimizer. L2 regularization was applied to prevent overfitting of the algorithm.

In the second stage, a neural network was designed accepting the clinical information with the features resulting from the first stage as input data. Specifically, the input variables were constructed by concatenating the five probabilities (probability to be KL grade 0~4 each), age, gender, BMI, DM, and HTN. The KL grade was finally predicted by applying 5-way Softmax activation function after going through 2 layers with 6 parameters each. In the second stage, the same weight initialization, learning rate, decay rate, decay step, and mini-batch size as the first stage were set. The same method as the first stage was used to minimize the loss function and prevent overfitting of the algorithm.

Class activation mapping (CAM) was used to find out how the DL algorithm detected KL stage. Class activation mapping was obtained by resizing the output to the input size using bilinear interpolation just before log-sum-exp pooling. Since KL grade was predicted by the 5-way Softmax activation function, class activation mappings for each KL grade could be obtained separately. In class activation mappings, Rectified Linear unit activation function (ReLU) was additionally applied to highlight the most sensitive regions in predicting KL grade.

## 3. Statistical Analysis

Sensitivity, specificity, and area under the receiver operating characteristic curve (AUC) were measured to evaluate the diagnostic performance of DL algorithm. An asymptotic calculation based on DeLong et al. [19] was used to obtain the 95% confidence interval for AUC. The AUC of the DL algorithm with sole image data and those of image and patient information were compared using the nonparametric test by DeLong et al. [19]. To calculate the sensitivity and specificity of the DL algorithm, three types of cutoffs were calculated based on the validation set: an optimal cutoff based on Youden’s J statistic was set, a cutoff with a sensitivity of 90%, and a cutoff with specificity of 90%.

All statistical analyses were performed using SPSS ver. 22.0 (IBM, Armonk, NY, USA). The data were presented as the means and standard deviations for continuous variables. The differences in the quantitative variables (i.e., age, BMI, and WBL ratio) were analyzed using Student’s t-test or Fisher exact test, as appropriate. Pearson chi-squared test or Fisher’s exact test were used to compare the qualitative variables (i.e., sex and patient’s history of DM and HTN). A *p* value less than 0.05 was considered statistically significant.

### 3.1. Results

The mean age and BMI of patients at the time of knee x-ray exams were 62.3 ± 12.8 years and 25.5 ± 3.19 kg/m^2^, respectively. The final inclusion of baseline characteristics of patients and distribution of labels in training, validation, and test sets is summarized in Table 1.

#### 3.1.1. Performance of DL with Sole Image Data

The AUCs of DL algorithm in diagnosing KL 0 to KL 4 with sole image data were as follows. In the validation set, the AUCs of DL algorithm in diagnosing KL 0 to KL 4 were 0.91 (95% confidence interval (CI), 0.87–0.95), 0.78 (95% CI, 0.72–0.84), 0.79 (95% CI, 0.74–0.84), 0.82 (95% CI, 0.78–0.86), and 0.95 (95% CI, 0.92–0.97), respectively. In the test set, the AUCs of DL algorithm in diagnosing KL 0 to KL 4 were 0.91 (95% CI, 0.88–0.95), 0.80 (95% CI, 0.76–0.84), 0.69 (95% CI, 0.64–0.73), 0.86 (95% CI, 0.83–0.89), and 0.96 (95% CI, 0.94–0.98), respectively (Figure 1A,B).

#### 3.1.2. Performance of DL with Image Data and Additional Patient Information

The performance of DL with image and additional patient data in diagnosing KL grade was competitive. In the validation set, the AUCs of DL algorithm in diagnosing KL 0 to KL 4 were 0.97 (95% confidence interval (CI), 0.95–0.98), 0.83 (95% CI, 0.78–0.87), 0.79 (95% CI, 0.75–0.84), 0.82 (95% CI, 0.78–0.87), and 0.95 (95% CI, 0.92–0.97), respectively. In the test set, the AUCs of DL algorithm in diagnosing KL 0 to KL 4 were 0.97 (95% confidence interval (CI), 0.71–0.74), 0.85 (95% CI, 0.80–0.86), 0.75 (95% CI, 0.66–0.73), 0.86 (95% CI, 0.79–0.85), and 0.95 (95% CI, 0.91–0.97), respectively (Figure 2A,B).

#### 3.1.3. Comparison of the Accuracy and Diagnosing Performance Between Sole Image Data and Image Data with Additional Patient Information

The diagnostic performance of image data with additional patient information had statistically significantly higher AUC than image data alone in diagnosing KL grades 0, 1, and 2 (*p*-values were 0.008, 0.020, and 0.027, respectively). The results are summarized in Table 2.

Higher sensitivity and specificity were observed in KL grades 0,1, and 2 of DL with the combination of image data and patient information. The sensitivity and specificity of the DL algorithm with image data alone and with combination of image data with additional patient information are listed in Table 3.

Diagnosing KL grade 2 was most challenging with the DL algorithm. Both analyses (one with image data alone and another with image data and additional patient information) tended to have difficulty in diagnosing KL grade 2 (Figure 3 and Figure 4).

#### 3.1.4. Gradient-Weighted Class Activation Mapping (Grad-CAM)

We applied the Grad-CAM technique to locate the most significant areas in the image for classification (Figure 5) [20].

They are key image data representing KL grades 0 to 4. The DL algorithm diagnosed all image data correctly in our study. For raw image data of KL grades 0, 3, and 4, the heat map signals appear on the knee joint area. Therefore, it appears that the DL algorithm and image data that we used detect the knee joint and can diagnose knee osteoarthritis without cropping image data. However, in the raw image data of KL 1, the heat map signal appeared only in the lateral side of the knee joint. In the raw image data of KL 2, the heat map signal did not appear on the knee joint. Thus, it would be complicated for the DL algorithm to detect and diagnose KL 1 and 2 with only raw image data.

## 4. Discussion

The principal findings of this study were as follows. DL was applied to diagnose knee OA on knee AP radiographs, and its performance was assessed with two distinct test sets. The diagnostic performance of DL was comparable to that of the former radiologist reading of knee OA KL grade; however, diagnosing KL 2 was most challenging in DL with or without additional patient information. Adding additional patient information to the image data increased the diagnostic performance (increased accuracy) of DL in diagnosing KL grades 0,1, and 2, with statistical significance. This is meaningful because our method could be used as an objective tool to support clinicians in their decision making of early OA.

Even the most commonly used KL grade scale is semi-quantitative and suffers from ambiguity. In the current study, the KL grading system was used to classify the severity of knee OA since it is most commonly used grading system. Based on this system, image data was labeled with the former radiologist KL grade reading. Then, using the DL algorithm, the accuracy of DL in diagnosing knee OA was tested. The performance of DL algorithm in diagnosing knee OA was comparable with other studies. Moreover, adding additional patient information improved the diagnostic performance of DL in knee OA. However, diagnosing KL grade 2 was challenging with or without additional patient information added. The AUCs of the DL algorithm in diagnosing KL 2 were 0.69 for image data alone and 0.75 for the combination of image data and additional patient information.

There were several studies that tried to enhance the performance and accuracy of making a diagnosis using DL algorithms [6,14,21]. To the best of our knowledge, most studies annotated the knee joint area in image data. Rather than converting the image data, the same image data used in the outpatient clinic was used. Additional patient data were added to the DL algorithm, just like a doctor sees a patient in the outpatient clinic. Therefore, it was possible to replicate a real outpatient clinical situation for diagnosing OA, determining the treatment modality, and predicting the prognosis.

Diagnosis of early OA is difficult in clinical practice for several reasons. It is well known that plain radiography is insensitive when attempting to detect early OA changes despite its advantages. This can be explained by several facts: first, a hallmark of OA and the best measure of its progression is the degeneration and wear of the articular cartilage—a tissue that cannot be directly seen in plain radiography; second, although the evaluation of the changes in the joint should be a three-dimensional problem, the image modality uses only a two-dimensional projection; and finally, the interpretation of the resulting image requires significantly-experienced radiologists. Eventually, the cartilage degeneration and wear are indirectly estimated by the assessment of joint-space narrowing and bony changes, like osteophytes and subchondral sclerosis [7]. Apart from the aforementioned limitations of plain radiography, OA diagnosis is also highly subjective due to the absence of precisely defined grading guidelines.

This study also has several limitations. First, the former reading of the KL grade was read by different radiologists, increasing the subjectivity of the reads. Second, we added limited patient information to the DL algorithm. Information regarding the physical examination of patients could also improve the diagnostic accuracy of DL. Third, the DL algorithm used in this study did not go through the process of localizing the knee joint. If this process was added before the KL grade prediction phase, an algorithm similar to the radiologist’s diagnosis could be designed. Fourth, even though the ratio of validation and test set is adequate, the number of test sets is small. In addition, the KL grade system’s ordinal properties were reflected in the loss function when designing the algorithm.

## 5. Conclusions

The diagnostic performance of DL was comparable to that of the former radiologist reading of the knee osteoarthritis KL grade. Additional patient information improved the diagnostic performance of DL for knee osteoarthritis.

## Figures and Tables

**Figure 1 jcm-09-03341-f001:**
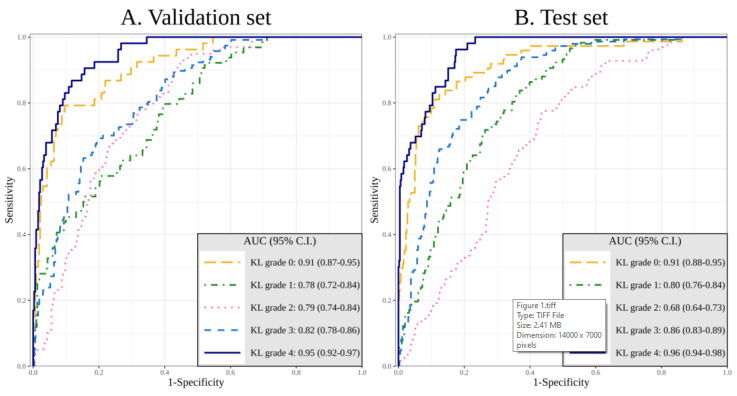
Area under the receiver operating curve (AUC) of the validation set (**A**) and the test set (**B**) for the deep learning (DL) algorithm of Kellgren–Lawrence (KL) grade 0 to 4 with sole image data.

**Figure 2 jcm-09-03341-f002:**
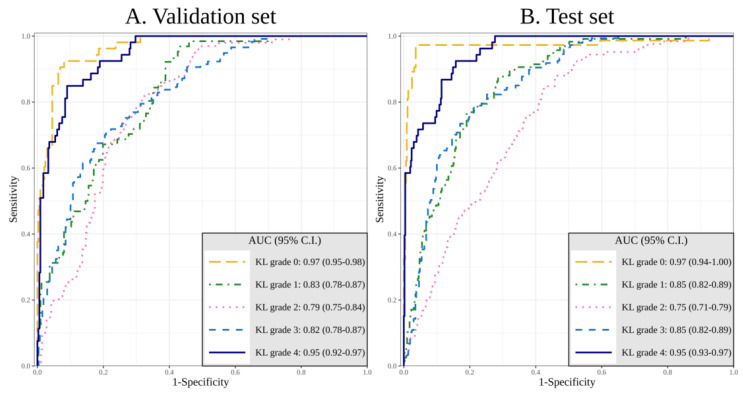
Area under the receiver operating curve of the validation set (**A**) and the test set (**B**) for the DL algorithm of KL grade 0 to 4 with image data and additional patient information.

**Figure 3 jcm-09-03341-f003:**
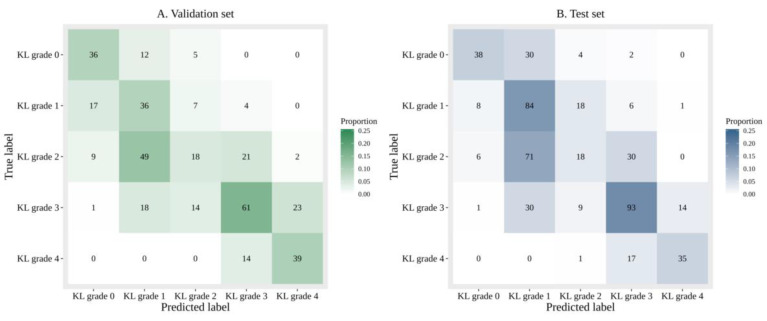
Confusion matrices of predicted and ground truth labels with sole image data.

**Figure 4 jcm-09-03341-f004:**
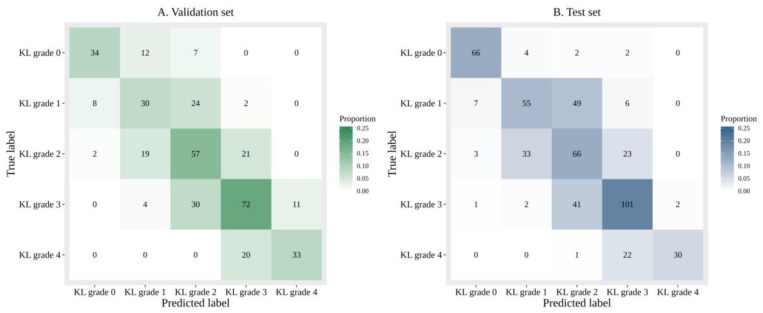
Confusion matrices of predicted and ground truth labels image data and additional patient information.

**Figure 5 jcm-09-03341-f005:**
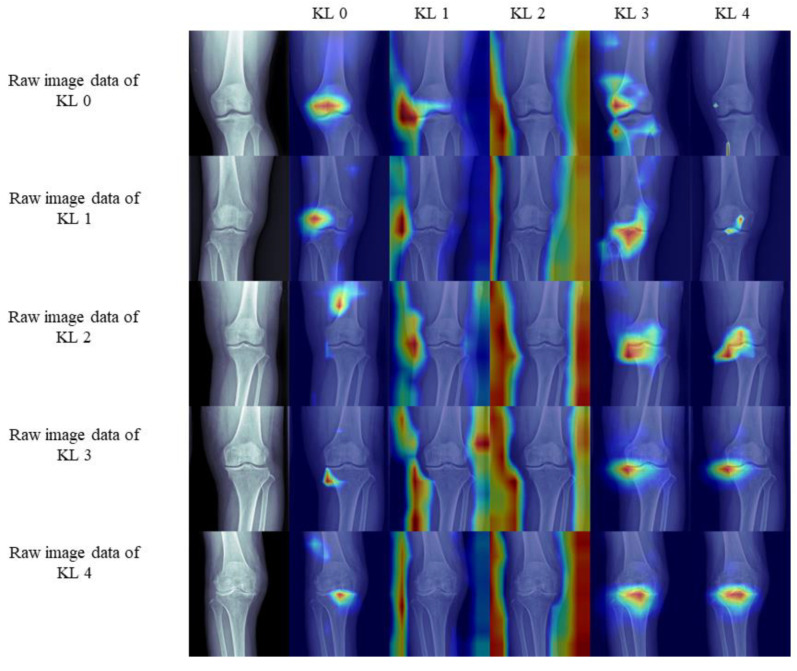
Qualitative results: Gradient-weighted class activation mapping.

**Table 1 jcm-09-03341-t001:** Baseline characteristics.

Characteristic	Training Set	Validation Set	Test Set	Total	*p*-Value
Age (Year)	62.3 ± 12.5	63.2 ± 12.6	61.5 ± 14.9	62.3 ± 2.8	0.359
Gender (M/F)	701/2763	74/312	146/370	921/3445	<0.001*
BMI (kg/m^2^)	25.5 ± 3.19	25.4 ± 2.96	25.9 ± 3.42	25.5 ± 3.20	0.025 *
WBL Ratio	0.31 ± 0.17	0.32 ± 0.15	0.36 ± 0.14	0.32 ± 0.16	<0.001 *
DM/HTN	629/1552	63/181	64/184	756/1917	0.005/<0.001 *
K-L 0	473	53	74	600	
K-L 1	574	64	117	755	
K-L 2	889	99	125	1113	
K-L 3	1055	117	147	1319	
K-L 4	473	53	53	579	
Total	3464	386	516	4366	

BMI: body mass index, WBL: weight bearing line, DM: diabetic mellitus, HTN: hypertension, K-L: Kellgren–Lawrence grade, *: statistically significant.

**Table 2 jcm-09-03341-t002:** Comparison of AUC of test sets.

	DL with Sole Image Data	DL with Image Data and Patient’s Information	*p*-Value
KL Grade 0	0.91 (0.88–0.95)	0.97 (0.94–1.00)	0.008 *
KL Grade 1	0.80 (0.76–0.84)	0.85 (0.82–0.89)	0.020 *
KL Grade 2	0.69 (0.64–0.73)	0.75 (0.71–0.79)	0.027 *
KL Grade 3	0.86 (0.83–0.89)	0.86 (0.82–0.89)	0.553
KL Grade 4	0.96 (0.94–0.98)	0.95 (0.93–0.97)	0.580

DL: deep learning, KL: Kellgren–Lawrence, *: statistically significant.

**Table 3 jcm-09-03341-t003:** Diagnostic performance of DL algorithm.

			Optimal Cutoff	Sensitivity of 90%	Specificity of 90%
**DL With** **Sole Image Data**	KL Grade 0	Sensitivity	64.9 (48/74) (52.9–75.6)	83.8 (2/74) (73.4–91.3)	64.9 (48/74) (52.9–75.6)
		Specificity	94.8 (419/442) (92.3–96.7)	84.2 (372/442) (80.4–87.4)	94.8 (419/442) (92.3–96.7)
	KL Grade 1	Sensitivity	94.0 (110/117) (88.1–97.6)	99.1 (116/117) (95.3–100.0)	58.1 (68/117) (48.6–67.2)
		Specificity	49.6 (198/399) (44.6–54.6)	38.6 (154/399) (33.8–43.6)	80.5 (321/399) (76.2–84.2)
	KL Grade 2	Sensitivity	80.0 (100/125) (71.9–86.6)	78.4 (98/125) (70.2–85.3)	29.6 (37/125) (21.8–38.4)
		Specificity	51.2 (200/391) (46.1–56.2)	51.9 (203/391) (46.8–57.0)	82.4 (322/391) (78.2–86.0)
	KL Grade 3	Sensitivity	75.5 (111/147) (67.7–82.2)	93.9 (138/147) (88.7–97.2)	56.5 (83/147) (48.0–64.6)
		Specificity	78.3 (289/369) (73.8–82.4)	61.5 (227/369) (56.3–66.5)	89.2 (329/369) (85.5–92.1)
	KL grade 4	Sensitivity	77.4 (41/53) (63.8–87.7)	84.9 (45/53) (72.4–93.3)	75.5 (40/53) (61.7–86.2)
		Specificity	90.7 (420/463) (87.7–93.2)	87.3 (404/463) (83.9–90.2)	92.0 (426/463) (89.2–94.3)
**DL With** **Image Data and** **Patient’s Information**	KL Grade 0	Sensitivity	97.3 (72/74) (90.6–99.7)	97.3 (72/74) (90.6–99.7)	97.3 (72/74) (90.6–99.7)
		Specificity	92.3 (408/442) (89.4–94.6)	93.9 (415/442) (91.2–95.9)	88.5 (391/442) (85.1–91.3)
	KL Grade 1	Sensitivity	92.3 (108/117) (85.9–96.4)	90.6 (106/117) (83.8–95.2)	48.7 (57/117) (39.4–58.1)
		Specificity	56.9 (227/399) (51.9–61.8)	62.2 (248/399) (57.2–66.9)	90.2 (360/399) (86.9–93.0)
	KL Grade 2	Sensitivity	76.0 (95/125) (67.5–83.2)	88.8 (111/125) (8.19–93.7)	27.2 (34/125) (19.6–35.9)
		Specificity	61.6 (241/391) (56.6–66.5)	51.4 (201/391) (46.3–56.5)	90.5 (354/391) (87.2–93.2)
	KL Grade 3	Sensitivity	73.5 (108/147) (65.6–80.4)	89.8 (132/147) (83.7–94.2)	51.0 (75/147) (42.7–59.3)
		Specificity	81.8 (302/369) (77.5–85.6)	61.5 (227/369) (56.3–66.5)	91.1 (336/369) (87.7–93.8)
	KL Grade 4	Sensitivity	73.6 (39/53) (59.7–84.7)	92.5 (49/53) (81.8–97.9)	73.6 (39/53) (59.7–84.7)
		Specificity	91.4 (423/463) (88.4–93.8)	84.2 (390/463) (80.6–87.4)	91.1 (422/463) (88.2–93.6)
**DL With** **Sole Image Data**	KL Grade 0	PPV	67.6 (48/71) (55.5–78.2)	47.6 (62/132) (38.2–55.8)	67.6 (48/71) (55.5–78.2)
		NPV	94.2 (419/445) (91.6–96.1)	96.9 (372/384) (94.6–98.4)	94.2 (419/445) (91.6–96.1)
	KL Grade 1	PPV	35.4 (110/311) (30.1–41.0)	32.1 (116/361) (27.3–37.2)	46.6 (68/146) (38.3–55.0)
		NPV	96.6 (198/205) (93.1–98.6)	99.4 (154/155) (96.5–100.0)	86.8 (321/370) (82.9–90.0)
	KL Grade 2	PPV	34.4 (100/291) (28.9–40.1)	34.3 (98/286) (28.8–40.1)	34.9 (37/106) (25.9–44.8)
		NPV	88.9 (200/225) (84.0–92.7)	88.3 (203/230) (83.4–92.1)	78.5 (322/410) (74.2–82.4)
	KL Grade 3	PPV	58.1 (111/191) (50.8–65.2)	49.3 (138/280) (43.3–55.3)	67.5 (83/123) (58.4–75.6)
		NPV	88.9 (289/325) (85.0–92.1)	96.2 (227/236) (92.9–98.2)	83.7 (329/393) (79.7–87.2)
	KL Grade 4	PPV	48.8 (41/84) (37.7–60.0)	43.3 (45/104) (33.6–53.3)	51.9 (40/77) (40.3–63.5)
		NPV	97.2 (420/432) (95.2–98.6)	98.1 (404/412) (96.2–99.2)	97.0 (426/439) (95.0–98.4)
	Accuracy		51.9 (268/516) (47.5–56.3)		
**DL With** **Image Data and** **Patient’s Information**	KL grade 0	PPV	67.9 (72/106) (58.2–76.7)	72.7 (72/99) (62.9–81.2)	58.5 (72/123) (49.3–67.3)
		NPV	99.5 (408/410) (98.2–99.9)	99.5 (415/417) (98.3–99.9)	99.5 (391/393) (98.2–99.9)
	KL grade 1	PPV	38.6 (108/280) (32.8–44.5)	41.2 (106/257) (35.2–47.5)	59.4 (57/96) (48.9–69.3)
		NPV	96.2 (227/236) (92.9–98.2)	95.8 (248/259) (92.5–97.9)	85.7 (360/420) (82.0–88.9)
	KL Grade 2	PPV	38.8 (95.245) (32.6–45.2)	36.9 (111/301) (31.4–42.6)	47.9 (34/71) (35.9–60.1)
		NPV	88.9 (241/271) (84.6–92.4)	93.5 (201/215) (89/3–96.4)	79.6 (354/445) (75.5–83.2)
	KL Grade 3	PPV	61.7 (108/175) (54.1–68.9)	48.2 (132/274) (42.1–54.3)	69.4 (75/108) (59.8–77.9)
		NPV	88.6 (302/341) (84.7–91.7)	93.8 (227/242) (90.0–96.5)	82.4 (336/408) (78.3–85.9)
	KL Grade 4	PPV	49.4 (39/79)	40.2 (49/122) (31.4–49.4)	48.8 (39/80) (37.4–60.2)
		NPV	96.8 (423/437) (94.7–98.2)	99.0 (390/394) (97.4–99.7)	96.8 (422/436) (94.7–98.2)
	Accuracy		61.6 (318/516) (57.3–65.8)		

DL: deep learning, KL: Kellgren–Lawrence, NPV: negative predictive value, PPV: positive predictive value.

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
