# Peer review of "Can Additional Patient Information Improve the Diagnostic Performance of Deep Learning for the Interpretation of Knee Osteoarthritis Severity"

_jcm, 2020, doi:10.3390/jcm9103341_

Round 1
Reviewer 1 Report
This is a well written paper that studies the capability of a DL algorithm to classify OA with clinical and radiologic data. I commend the authors for exploring a current subject of interest. However, there are several issues with this paper I would like them to address:
- Why would you limit the clinical information to age, gender, BMI and WBL? There are lots of more parameters, that a human physician considers when evaluating OA, especially for indication of surgery, like the dosage and potency of pain medication the patient has been taking, the duration of failed conservative treatment, joint pain when standing up from sitting position or pain in the morning, pain intensity as measured in a VAS scale, etc. Please justify.
- A training set in a 9:1 ratio is a HUGE training set. Usually, the training set should not exceed 50% of samples. Why did you choose such a big training set? My guess would be that your algorithm's performance would considerably decline with a bigger test set? Please justify.
- Please justify how and why you chose a learning rate, decay rate, and decay step of 117 0.01, 0.94, and 5,000, I guess you used the validation set for the optimal parameter selection?
- Your descprition of your image clasification algorithm of the X-ray images is too general. I would like to understand how your DL algorithm ended up classifying the images into the KL scala. Did you use any filters to strengthen the contours? You mention grad-CAM in the Results section but do not describe it in the Methods section. How did you select the features? What were the features that finally allowed a classification of each single KL stage?
- -When you claim "All DICOM files were matched with radiologists’ former reading of KL grade." do you mean you used as output comparison for your DL algorithm the pre-existing KL grading? Was this initial grading performed by a single human reader or several readers? Please reason if this could be a limitation regarding the generalization of your algorithm regarding to intra- and inter-observer variability.
- I still do not understand how you selected the final feature set with which you feeded your algorithm. Did you use any feature selection algorithm like PCA, Adaboost, etc.? A diagram depicting the steps of your algorithm's feature selection, training, validation and testing displaying the most important parameters would help the reader better understand your steps.
- It would be helpful if the authors would also list NPV, PPV and accuracy values for their algorithm with the 95% CI values for all parameters (AUC, Se, Spec, etc).
Author Response
Response to Reviewer 1 Comments
This is a well written paper that studies the capability of a DL algorithm to classify OA with clinical and radiologic data. I commend the authors for exploring a current subject of interest. However, there are several issues with this paper I would like them to address:
Point 1: Why would you limit the clinical information to age, gender, BMI and WBL? There are lots of more parameters, that a human physician considers when evaluating OA, especially for indication of surgery, like the dosage and potency of pain medication the patient has been taking, the duration of failed conservative treatment, joint pain when standing up from sitting position or pain in the morning, pain intensity as measured in a VAS scale, etc. Please justify. 

Response 1: Thank you for your valuable comments. We totally agree with your comments. We had to limit the clinical information because the data we treated was too large. This is the limitation of our study. Therefore, it is mentioned in the manuscript line 245-247. We could improve the diagnostic accuracy of DL if we could include other parameters of patient clinical information.
Point 2: A training set in a 9:1 ratio is a HUGE training set. Usually, the training set should not exceed 50% of samples. Why did you choose such a big training set? My guess would be that your algorithm's performance would considerably decline with a bigger test set? Please justify.
Response 2: Thank you for your valuable comments. Usually in deep learning studies, the ratio of the training set is adjusted by 50% or more. In our study, the training set and validation wet were divided at a 9:1 ratio to secure more training set for higher performance of our algorithm.
Point 3: Please justify how and why you chose a learning rate, decay rate, and decay step of 117 0.01, 0.94, and 5,000, I guess you used the validation set for the optimal parameter selection?
Response 3: Thank you for your valuable comments. The learning rate was set to a value that the training rate was not too slow or overfitting. The decay step was determined in proportion to the number of training sets. The decay rate was set to a value that was generally used. The learning rate, decay rate, and decay step were set to control the learning rate during training, and optical parameters were obtained based on the average value of the AUCs of each KL grade obtained from the validation set.
Point 4: Your description of your image classification algorithm of the X-ray images is too general. I would like to understand how your DL algorithm ended up classifying the images into the KL scale. Did you use any filters to strengthen the contours? You mention grad-CAM in the Results section but do not describe it in the Methods section. How did you select the features? What were the features that finally allowed a classification of each single KL stage?
Response 4: Thank you for your valuable comments. We did not use any filters to strengthen the contours. The process of obtaining CAM was additionally described in the method section. (Line 128-133) The area close to the red color in the Cam is the most sensitive area to predict the KL grade.
Point 5: When you claim "All DICOM files were matched with radiologists’ former reading of KL grade." do you mean you used as output comparison for your DL algorithm the pre-existing KL grading? Was this initial grading performed by a single human reader or several readers? Please reason if this could be a limitation regarding the generalization of your algorithm regarding to intra- and inter-observer variability.
Response 5: Thank you for your valuable comments. Yes, we used output comparison for our DL algorithm the pre-existing KL grading. The initial grading of KL grade was performed by several radiologists in our hospital. We thought this could be our limitation because it increases the subjectivity of KL grade reading. This is mentioned in Line 244-245.
Point 6: I still do not understand how you selected the final feature set with which you feeded your algorithm. Did you use any feature selection algorithm like PCA, Adaboost, etc.? A diagram depicting the steps of your algorithm's feature selection, training, validation and testing displaying the most important parameters would help the reader better understand your steps.
Response 6: Thank you for your valuable comments. Algorithms like ADAboost were not used. The specific process of confirming the final feature set using class activation mapping is additionally described in the method section. (Line 128-133)
Point 7: It would be helpful if the authors would also list NPV, PPV and accuracy values for their algorithm with the 95% CI values for all parameters (AUC, Se, Spec, etc).
Response 7: Thank you for your valuable comments. We added the values of NPV, PPV and accuracy values for our algorithm with the 95% CI values for all parameters. (Table 3.)
Reviewer 2 Report
the article is well written and innovative for the chosen topic, images and tables are very explanatory of the results obtained by the authors. I suggest implementing the introduction, considering also other factors that may intervene in the problem under investigation (example: "The effect of a functional appliance in the management of temporomandibular joint disorders in patients with juvenile idiopathic arthritis.
Island G, et al. Minerva Stomatol. 2017. PMID: 27716739 ").
Author Response
Response to Reviewer 2 Comments
Point 1: the article is well written and innovative for the chosen topic, images and tables are very explanatory of the results obtained by the authors. I suggest implementing the introduction, considering also other factors that may intervene in the problem under investigation (example: "The effect of a functional appliance in the management of temporomandibular joint disorders in patients with juvenile idiopathic arthritis.
Island G, et al. Minerva Stomatol. 2017. PMID: 27716739 ").
Response 1: Thank you for your valuable comments. We have mentioned the examples of the strength of DL in image classification. (Line 44-49)
Round 2
Reviewer 1 Report
The authors have improved the quality of the manuscript with the corrections.
However, I am not satisfied with the following answer.
"Response 2: Thank you for your valuable comments. Usually in deep learning studies, the ratio of the training set is adjusted by 50% or more. In our study, the training set and validation wet were divided at a 9:1 ratio to secure more training set for higher performance of our algorithm."
Of course your algorithm improves with an extensive training set (if you would use 99% of your dataset as a training, your algorithm would perform even better on the remaining test set). Therefore, my concern is that with such a huge training set, your current test set is too small to prove the performance of your algorithm. A good classification algorithm needs to be tested with an extensive test set, otherwise its generalization capability is limited.
Please try to justify again why your test set was so small (or your training set needed to be that large) and you should add this to the limitation section of your manuscript.
Author Response
Response to Reviewer 1 Comments
Point 1: The authors have improved the quality of the manuscript with the corrections.
However, I am not satisfied with the following answer.
"Response 2: Thank you for your valuable comments. Usually in deep learning studies, the ratio of the training set is adjusted by 50% or more. In our study, the training set and validation wet were divided at a 9:1 ratio to secure more training set for higher performance of our algorithm."
Of course your algorithm improves with an extensive training set (if you would use 99% of your dataset as a training, your algorithm would perform even better on the remaining test set). Therefore, my concern is that with such a huge training set, your current test set is too small to prove the performance of your algorithm. A good classification algorithm needs to be tested with an extensive test set, otherwise its generalization capability is limited.
Please try to justify again why your test set was so small (or your training set needed to be that large) and you should add this to the limitation section of your manuscript.
Response 1: Thank you for your valuable comments. We understand your concern, however in our study we divided development set similar to other DL studies. The reference is added in the manuscript. (Line 83-84, Reference 11, 16) Even though the ratio of validation set and test set is adequate, the number of our test set is small. We added this matter the limitation section as you suggested. (Line 250-251)